# Association between Frequency of Central Respiratory Events and Clinical Outcomes in Heart Failure Patients with Sleep Apnea

**DOI:** 10.3390/jcm11092403

**Published:** 2022-04-25

**Authors:** Ryo Naito, Takatoshi Kasai, Koji Narui, Shin-Ichi Momomura

**Affiliations:** 1Department of Cardiovascular Biology and Medicine, Graduate School of Medicine, Juntendo University, Tokyo 113-8421, Japan; rnaitou@juntendo.ac.jp; 2Cardiovascular Respiratory Sleep Medicine, Graduate School of Medicine, Juntendo University, Tokyo 113-8421, Japan; 3Sleep Center, Toranomon Hospital, Tokyo 105-8470, Japan; k.narui.bv@juntendo.ac.jp; 4Department of Medicine, Saitama Citizens Medical Center, Saitama 331-0054, Japan; momoshin@omiya.jichi.ac.jp

**Keywords:** heart failure, central sleep apnea, cardiovascular events

## Abstract

Heart failure (HF) is a progressive cardiac disorder associated with high mortality and morbidity. Previous studies have shown that sleep apnea (SA) is associated with a poor prognosis in HF patients. When HF coexists with SA, both central and obstructive respiratory events often occur. However, few studies have investigated the association between the frequency of central respiratory events coexisting with obstructive events and clinical outcomes in patients with HF and SA. This was a retrospective observational study. Patients with stable HF, defined as a left ventricular ejection fraction of ≤50%, New York Heart Association class ≥ II, and SA (apnea–hypopnea index of ≥15/h on overnight polysomnography) were enrolled. The primary endpoint was a composite of all-cause death and hospitalization for HF. Overall, 144 patients were enrolled. During a period of 23.4 ± 16 months, 45.8% of patients experienced the outcome. The cumulative event-free survival rates were higher in the central SA-predominant group. Multivariate analyses showed that a greater percentage of central respiratory events was associated with an increased risk of clinical outcomes. In patients with HF and SA, the frequency of central respiratory events was an independent factor for all-cause death and hospitalization for HF.

## 1. Introduction

Heart failure (HF) is an advanced stage of cardiac disease and is associated with a high rate of mortality and hospitalization due to the exacerbation of HF. The exacerbation occurs despite providing evidence-based medical and non-pharmacological therapies [1,2]. The identification of factors that contribute to an increased risk of mortality could aid in improving survival rates in HF patients. Several studies have reported that sleep apnea (SA), regardless of the type, namely, central SA (CSA) or obstructive SA (OSA), is associated with a poor prognosis in HF patients [3,4]. CSA in HF patients occurs because of increased reactivity of central chemoreceptors of carbon dioxide, repeated hypo- and hyperventilation, and extended circulation time [4,5]. Often, CSA and OSA exist in HF patients. However, only a few studies have examined the association between the percentage of central to total respiratory events and prognosis in HF patients. We hypothesized that a higher frequency of CSA is associated with worse clinical outcomes in HF patients because a greater number of central respiratory (central apnea or hypopnea) events might occur under severe HF conditions [6]. Therefore, this study aimed to examine the association between the percentage of central apnea or hypopnea events to total events (% C/T) and prognosis in patients with HF and SA.

## 2. Materials and Methods

### 2.1. Study Population

Consecutive patients with HF and moderate to severe SA who participated in the follow-up conducted at the cardiovascular center of Toranomon Hospital (Tokyo, Japan) between 1 January 2001 and 1 March 2005, were enrolled in the study. 

The inclusion criteria were as follows: (1) presence of symptomatic HF, defined as a left ventricular ejection fraction (LVEF) of ≤50% on echocardiography within one month before the diagnostic sleep study and New York Heart Association (NYHA) class II or above; (2) stable clinical status, defined as no hospital admissions within one month before study enrollment and receiving optimal medical therapy for at least one month before study enrollment; and (3) having undergone a sleep study and received a diagnosis of moderate-to-severe SA, which was defined as ≥15 apnea or hypopnea events per hour of sleep (i.e., apnea–hypopnea index [AHI]). Respiratory events were scored according to the updated American Academy of Sleep Medicine (AASM) 2020 guideline [7]. The exclusion criteria were as follows: (1) age below 20 or above 80 years, (2) presence of known untreated neoplasms, (3) history of stroke with neurologic deficits, and (4) history of severe chronic pulmonary diseases. All eligible patients were classified into two groups according to the median value of % C/T (predominant CSA group, % C/T ≥ 30.1%; non-predominant CSA group, % C/T < 30.1%). Informed consent was obtained orally from all study participants. The study was conducted in compliance with the Declaration of Helsinki and in accordance with the ethics policies of the involved institutions. 

### 2.2. Respiratory Event Estimation

Respiratory events (i.e., apneas or hypopneas) were scored according to the 2020 AASM scoring manual updates. Apnea, with and without ribcage and/or abdominal movement, was defined as obstructive and central apnea, respectively. Hypopnea was classified as obstructive if any of the following conditions existed: (1) paradoxical chest and abdominal movement during hypopnea events, (2) snoring during hypopnea events, or (3) flow limitation on the nasal pressure signals. Otherwise, the hypopneas were classified as central. 

### 2.3. Outcome Measure

Patients’ characteristics, polysomnography data, and clinical outcomes were assessed. Clinical outcomes, defined as a composite of all-cause death and hospitalization for HF until March 2006, were compared between the two groups.

### 2.4. Statistical Analysis

The data of all variables were presented as mean ± standard deviation. The baseline characteristics were compared using Student’s *t*-test or the Mann–Whitney U test for continuous variables, while the χ^2^ test or Fisher’s exact test were used for categorical variables. Event-free survival between the groups was compared using the Kaplan–Meier estimate with the log-rank test, and hazard ratios (HRs) were calculated using the Cox proportional hazards model. The assumption of proportional hazards was assessed by plotting a log-minus-log survival graph. Univariate analysis was based on the proportional hazards model to determine the associations between prognosis and the following pretreatment variables between the groups: age, sex, body mass index, LVEF, brain natriuretic peptide (BNP) levels, plasma norepinephrine concentration, NYHA class, systolic/diastolic blood pressure, heart rate, HF etiology, atrial fibrillation, Epworth sleepiness score, medications, total AHI, percentage of time spent at oxygen saturation (SpO_2_) below 90%, lowest SpO_2_, ratio of slow-wave sleep or rapid eye movement sleep to total sleep time, continuous positive airway pressure (CPAP) therapy, and % C/T as a categorical variable (% C/T cutoff value = 30.1%) or % C/T as a continuous variable. Variables with a *p* value below 0.1 in univariate analysis were included in the multivariate analysis. Statistical significance was set at *p* < 0.05. All statistical analyses were performed using a statistical software package (SPSS, version 11.0 for Windows; SPSS Inc., Chicago, IL, USA).

## 3. Results

A total of 144 patients were enrolled in this study. The median follow-up period was 23.4 ± 16.0 months. The baseline characteristics of the patients are shown in Table 1. BNP level, NYHA class, and prevalence of atrial fibrillation were significantly higher in the predominant CSA group. Additionally, there was a tendency toward higher age in that group. Medication use was comparable between the two groups (Table 2). Polysomnographic data showed that the total AHI was higher in the predominant CSA group than in the non-predominant group (46.3 ± 17.8 and 40.4 ± 17.9, respectively; *p* = 0.049). Mean % C/T was 71.3 ± 22.9% in the predominant CSA group and 10.9 ± 13.5% in the counterpart (*p* < 0.001; Table 3). Kaplan–Meier estimation of cumulative event-free survival revealed a significantly worse clinical outcome in the predominant CSA group (log-rank test *p* = 0.001) (Figure 1). A multivariate Cox proportional hazard model was used for the primary endpoint for both groups. Model 1, in which % C/T was a categorical variable, showed that % C/T ≥ 30.1%, BNP, and atrial fibrillation were associated with increased risk for clinical outcomes, while systolic blood pressure, LVEF and CPAP therapy were associated with reduced risk for the outcomes. Model 2, in which % C/T was treated as a continuous variable, showed a result similar to that of Model 1. The degree of % C/T as a categorical variable and as a continuous variable was found to be associated with worse clinical outcomes in patients with HF and SA (Table 4).

In Model 1, the following variables were included as covariates: age, sex, systolic BP, LVEF, BNP, atrial fibrillation, beta blocker use, and CPAP therapy, in addition to % C/T of ≥30.1% (categorical variable). In Model 2, the aforementioned variables were included as covariates. Additionally, instead of % C/T of ≥30.1% (categorical variable), % C/T (continuous variable) was included.

## 4. Discussion

In this study, we demonstrated that a higher % C/T was associated with worse clinical outcomes in HF patients with SA. Previous studies reported that SA is a negative predictor in HF patients [4,8,9]. Recurrent episodes of apneas during sleep, followed by arousals, generate repetitive hypoxia-reoxygenation, increased intrathoracic negative pressure, and exaggerated sympathetic nervous activity, all of which contribute to an increased ventricular afterload [3]. CSA is known to occur secondary to HF through increased reactivity of central chemoreceptors of carbon dioxide, repeated hypo- and hyperventilation, and extended circulation time accompanied by reduced cardiac output [4]. Regarding clinical outcomes in HF patients complicated with CSA, Javaheri et al. compared the mortality of HF patients with and without CSA. They reported that the median survival period of patients with CSA was significantly shorter than that of patients without CSA (HR 2.14, *p* = 0.02) [10]. CSA was found to be a predictor after controlling for confounding variables in that study [10]. In patients with HF, both CSA and OSA often exist, and more central respiratory events may occur under severe HF conditions [6]. However, the association between CSA frequency and prognosis in patients with HF and SA has not been elucidated. Therefore, to the best of our knowledge, our study is the first to report the finding that a higher % C/T was associated with worse clinical outcomes in patients with HF and SA. To date, it remains undetermined whether CSA simply reflects severely impaired cardiac function with pulmonary congestion or whether CSA independently affects prognosis in HF patients through several mechanisms [11,12]. Repetitive hypoxia-reoxygenation induced by apneas, hypopneas, and arousals can induce excessive sympathetic nervous activation [13]. Moreover, a prior study reported that a subset of patients with CSA and hyperpnea had longer hyperpnea and higher N-terminal pro BNP levels, indicating reduced cardiac output and greater tension against the left ventricular wall, which may consequently lead to a higher mortality rate [14]. The underlying mechanism that explains why a greater degree of % C/T is associated with worse clinical outcomes is unknown. We speculated that disadvantages in the patient profile in the predominant CSA group, wherein the proportion of elderly patients with atrial fibrillation was higher and the severity of HF was greater considering the greater NYHA class and higher BNP value, affected prognosis in these patients, although % C/T remained a predictor after controlling for confounding variables.

### 4.1. Clinical Implications and Future Perspectives

In this study, we did not elucidate whether % C/T could be an interventional target to improve prognosis in patients with HF and SA. However, CPAP therapy was proven to be associated with reduced risk for a composite of all-cause death and hospitalization for HF, although follow-up polysomnographic data after the initiation of CPAP therapy and treatment compliance to CPAP were not available. CPAP improves cardiac function in patients with HF and CSA [15,16,17]. Post-hoc analysis of the CANPAP (Canadian Continuous Positive Airway Pressure for Patients with Central Sleep Apnea and Heart Failure) trial showed improvement in terms of transplant-free survival and LVEF in patients whose CSA was suppressed by CPAP [18]. However, CPAP was originally not prepared to suppress CSA, and previous studies reported that some individuals are non-responders to CPAP. Recently, adaptive servo-ventilation (ASV) has been reported to be more effective than CPAP in suppressing CSA in patients with HF and CSA [19,20]. However, the SERVE-HF trial, in which 1325 patients with HF and predominant CSA (% C/T > 50%) were randomly assigned to guideline-based medical therapy with ASV or guideline-based medical therapy alone, demonstrated that all-cause and cardiovascular mortality rates were higher in the ASV group than in the medical therapy alone group [21]. Although the reason for the increased mortality in the ASV group is not known, several potential mechanisms were suggested in terms of ASV-induced positive intrathoracic pressure that can lower cardiac output, inspiratory pressure support that may lead to hyperventilation, alkalosis, and accompanying hypokalemia that could induce cardiac arrhythmias, and the inclusion of participants with a % C/T m > 50%. To date, no study has investigated the efficacy of ASV in patients with a % C/T of >30%. Further studies are needed to examine the cutoff value of % C/T as a prognostic factor in patients with HF and CSA.

### 4.2. Limitations

This was a single-center observational study with a relatively small sample size. Our data should be interpreted carefully, and further studies with larger populations are required to confirm our data.

## 5. Conclusions

Our study demonstrated that patients who had HF and a higher frequency of CSA had a higher risk of all-cause death and hospitalization for HF.

## Figures and Tables

**Figure 1 jcm-11-02403-f001:**
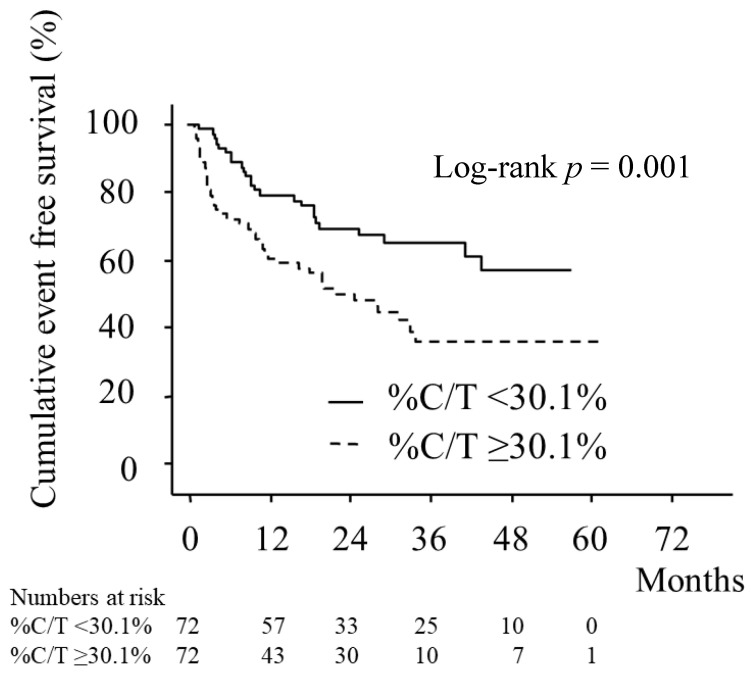
Kaplan–Meier estimation of event-free survival. Legend: Kaplan–Meier estimation of cumulative event-free survival showing a significantly worse clinical outcome in the predominant central sleep apnea group. % C/T, percentage of central to total respiratory events.

**Table 1 jcm-11-02403-t001:** Baseline characteristics of patients in the two groups.

Characteristic	% C/T ≥ 30.1% (*n* = 72)	% C/T < 30.1% (*n* = 72)	*p*
Age, years	64.3 ± 12.9	60.4 ± 12.8	0.076
Male sex, *n* (%)	65 (90.3)	66 (91.7)	0.771
BMI, kg/m^2^	25.7 ± 6.3	26.8 ± 5.2	0.266
Systolic BP, mmHg	130.9 ± 9.8	130.9 ± 12.6	0.965
Diastolic BP, mmHg	77.3 ± 10.9	79.1 ± 9.8	0.309
HR, /min	75.5 ± 8.6	76.5 ± 11.1	0.519
LVEF, %	36.3 ± 9.6	38.9 ± 9.4	0.101
BNP, pg/mL	440.1 ± 56.2	256.6 ± 24.1	0.003
PNE, pg/mL	555.1 ± 25.0	515.6 ± 17.2	0.196
NYHA class			0.030
II, *n* (%)	30 (41.7)	45 (62.5)	
III, *n* (%)	38 (52.8)	26 (36.1)	
IV, *n* (%)	4 (5.5)	1 (1.4)	
Ischemic, *n* (%)	23 (31.9)	17 (23.6)	0.264
Atrial fibrillation, *n* (%)	38 (52.8)	24 (33.3)	0.019
ICD, *n* (%)	4 (5.6)	1 (1.4)	0.172
ESS	8.7 ± 3.9	9.1 ± 3.9	0.493
CPAP therapy, *n* (%)	57 (78.2)	54 (75.0)	0.552

% C/T, percentage of central to total respiratory events, BMI, body mass index; BP, blood pressure; HR, heart rate; LVEF, left ventricular ejection fraction; BNP, brain natriuretic peptide; PNE, plasma norepinephrine; NYHA, New York Heart Association; ICD, implantable cardioverter defibrillator; ESS, Epworth Sleepiness Scale; CPAP, positive airway pressure.

**Table 2 jcm-11-02403-t002:** Medication use.

Medication	% C/T ≥ 30.1% (*n* = 72)	% C/T < 30.1% (*n* = 72)	*p*
Beta-blockers, *n* (%)	47 (65.3)	38 (52.8)	0.127
ACE inhibitors/ARBs, *n* (%)	62 (86.1)	61 (84.7)	0.813
Diuretics, *n* (%)	58 (80.6)	57 (79.2)	0.835
Digoxin, *n* (%)	26 (36.1)	23 (31.9)	0.598

% C/T, percentage of central to total respiratory events, ACE, angiotensin-converting enzyme; ARB, angiotensin II receptor blocker; % C/T, percentage of central to total respiratory events.

**Table 3 jcm-11-02403-t003:** Polysomnographic findings.

	% C/T ≥ 30.1% (*n* = 72)	% C/T < 30.1% (*n* = 72)	*p*
TST, min	317.7 ± 76.3	322.5 ± 84.0	0.730
Total AHI, /h	46.3 ± 17.8	40.4 ± 17.9	0.049
Obstructive AHI, /h	14.5 ± 13.8	36.2 ± 16.4	<0.001
Central AHI, /h	31.8 ± 14.6	4.2 ± 4.9	<0.001
% C/T, %	71.3 ± 22.9	10.9 ± 13.5	<0.001
% TST SpO_2_ < 90%, %	30.6 ± 31.2	25.1 ± 30.5	0.290
Lowest SpO_2,_ %	78.4 ± 9.1	75.4 ± 15.2	0.157
Arousal index, /h	40.5 ± 21.5	39.1 ± 19.5	0.692
Sleep stage, % of TST			
Slow wave sleep, %	5.6 ± 7.9	8.1 ± 8.7	0.081
REM sleep, %	10.2 ± 7.2	9.8 ± 6.5	0.734

% C/T, percentage of central to total respiratory events, TST, total sleep time; AHI, apnea–hypopnea index; % C/T, percentage of central to total respiratory events; SpO_2_, oxyhemoglobin saturation; REM, rapid eye movement.

**Table 4 jcm-11-02403-t004:** Results of multivariate analysis assessing prognostic factors for clinical outcomes.

Factors		Model 1		Model 2		
	HR	95% CI	*p*	HR	95% CI	*p*
Age (1-year increase)	1.01	0.99–1.03	0.397	1.01	0.99–1.03	0.518
Male sex	1.44	0.62–3.34	0.396	1.68	0.70–4.00	0.245
Systolic BP (1-mmHg increase)	0.98	0.94–0.99	0.013	0.97	0.94–0.99	0.026
LVEF (1% increase)	0.97	0.95–0.99	0.035	0.97	0.94–0.99	0.012
BNP (10-pg/mL increase)	1.06	1.02–1.09	0.003	1.05	1.01–1.09	0.007
Atrial fibrillation	1.87	1.15–3.05	0.012	1.75	1.08–2.82	0.023
Beta blocker use	0.67	0.39–0.95	0.045	0.63	0.37–1.10	0.105
CPAP therapy	0.50	0.29–0.86	0.012	0.51	0.30–0.87	0.014
% C/T ≥ 30.1%	2.16	1.27–3.68	0.005	-		
% C/T (1% increase)	-			1.02	1.01–1.03	<0.001

BNP, brain natriuretic peptide; BP, blood pressure; CI, confidence interval; HR, hazard ratio; LVEF, left ventricular ejection fraction; CPAP, positive airway pressure; % C/T, percentage of central to total respiratory events.

## Data Availability

Deidentified participant data will be shared on request from the corresponding author.

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
