# Peer review of "Association between Frequency of Central Respiratory Events and Clinical Outcomes in Heart Failure Patients with Sleep Apnea"

_jcm, 2022, doi:10.3390/jcm11092403_

Round 1

Reviewer 1 Report

The object of this study was the occurrence of readmission to hospital or death during follow-up among patients with heart failure and sleep apnea observed between 2001 and 2006, in relationship to the proportion of events of the central type during sleep. The authors retrospectively evaluated medical records and polysomnography data, scored with the most modern criteria. The most important finding of this study was that percentage of central over total respiratory events (C/T) was associated with worse outcomes both if considered as a categorical (below or above the median value of 30.1%) and as a continuous variable, even after adjustment for a high number of possible confounders. This finding confirms previous observations in the literature that showed that in heart failure central apneas are markers of a more severe disease In addition to C/T, other predictors of the outcome were identified.

The data are reported clearly and in detail. Some aspects of the discussion appear questionable. English language should be improved.

  1. Abstract, lines 22-23. This sentence should be improved.
  2. Lines 47-48 and 170. The main chemoreceptors for carbon dioxide are central, not pulmonary. Besides, the term “level” here must be replaced (e.g., reactivity or responsiveness).
  3. Line 54-55. The authors did not address exactly the frequency of central events (the central apnea/hypopnea index), but the percentage of the central on the total events (C/T). I suggest to clarify.
  4. Were consecutive unselected patients meeting inclusion criteria enrolled?
  5. Lines 75-77. It should be clarified whether informed consent was given or not.
  6. Lines 125-126. It would be useful to specify which of these variables was a positive, and which one was a negative, predictor of the outcome.
  7. Did the authors know what type of PAP was used? Was any patient prescribed adaptive servo ventilation (ASV)? Besides, effects of PAP may differ between patients with central and obstructive apneas. If feasible, it could be useful to assess them separately in the two groups.
  8. Lines 185-187. I suggest to remodulate this sentence. The effects of hyperpnea following central events on intrathoracic pressure and cardiac output are not as simple as they are described in the manuscript (see, e.g., Perger E et al, J Clin Sleep Med 2017). Ref 14, reported here, does not support what the authors state. Besides, intrathoracic pressure changes during obstructive apneas can cause significant reductions in stroke volume, and I am not sure that they are less marked than those that may be observed during hyperpnea following central events.

Author Response

Response to Reviewers’ comments

We fully appreciate editor’s and 3 reviewers’ kind review and comments for our work. We attempted to incorporate them into the revised manuscript. We believe that the revision has been improved sufficiently enough for publication of your journal.

Following are responses to specific comments from the Reviewer 1.

Reviewer 1

The object of this study was the occurrence of readmission to hospital or death during follow-up among patients with heart failure and sleep apnea observed between 2001 and 2006, in relationship to the proportion of events of the central type during sleep. The authors retrospectively evaluated medical records and polysomnography data, scored with the most modern criteria. The most important finding of this study was that percentage of central over total respiratory events (C/T) was associated with worse outcomes both if considered as a categorical (below or above the median value of 30.1%) and as a continuous variable, even after adjustment for a high number of possible confounders. This finding confirms previous observations in the literature that showed that in heart failure central apneas are markers of a more severe disease In addition to C/T, other predictors of the outcome were identified.

The data are reported clearly and in detail. Some aspects of the discussion appear questionable. English language should be improved.

Specific comments:

Abstract, lines 22-23. This sentence should be improved.

Response to the comment

The sentence is revised as follows “Heart failure (HF) is a progressive cardiac disorder associated with high mortality and morbidity”.

Lines 47-48 and 170. The main chemoreceptors for carbon dioxide are central, not pulmonary. Besides, the term “level” here must be replaced (e.g., reactivity or responsiveness).

Response to the comment

Revision was made according to the comment.

Line 54-55. The authors did not address exactly the frequency of central events (the central apnea/hypopnea index), but the percentage of the central on the total events (C/T). I suggest to clarify.

Response to the comment

The central events include central apnea or hypopnea events. The sentence was revised.

Were consecutive unselected patients meeting inclusion criteria enrolled?

Response to the comment

Yes. Revision was made in Study population section.

Lines 75-77. It should be clarified whether informed consent was given or not.

Response to the comment

As described in the original version, informed consent was obtained orally from all study participants.

Lines 125-126. It would be useful to specify which of these variables was a positive, and which one was a negative, predictor of the outcome.

Response to the comment

Revision was made on the sentences as follows, “Model 1, in which % C/T was a categorical variable, showed that % C/T≥30.1%, BNP, and atrial fibrillation were associated with increased risk for clinical outcomes while systolic blood pressure, LVEF and PAP therapy were associated with reduced risk for the outcomes. Model 2, in which % C/T was treated as a continuous variable, showed a result similar to that of Model 1”.

Did the authors know what type of PAP was used? Was any patient prescribed adaptive servo ventilation (ASV)? Besides, effects of PAP may differ between patients with central and obstructive apneas. If feasible, it could be useful to assess them separately in the two groups.

Response to the comment

CPAP was the only PAP therapy used for the study participants. PAP was replaced with CPAP in the revision.

Lines 185-187. I suggest to remodulate this sentence. The effects of hyperpnea following central events on intrathoracic pressure and cardiac output are not as simple as they are described in the manuscript (see, e.g., Perger E et al, J Clin Sleep Med 2017). Ref 14, reported here, does not support what the authors state. Besides, intrathoracic pressure changes during obstructive apneas can cause significant reductions in stroke volume, and I am not sure that they are less marked than those that may be observed during hyperpnea following central events.

Response to the comment

The part was revised according to the comment from “Moreover, exaggerated alteration of intrathoracic pressure during the hyperpnea phase following central apnea could be associated with reduced cardiac output, which leads to a higher mortality rate in HF patients” to “Moreover, a prior study reported that a subset of patients with CSA and hyperpnea had longer hyperpnea and higher N-terminal pro BNP levels, indicating reduced cardiac output and greater tension against left ventricular wall, which may consequently lead to a higher mortality rate14” (Lines 194-198).The Ref 14 in the original draft was replaced with the suggested article.

Reviewer 2 Report

In this study the researchers have examined whether different types of sleep apnea have different clinical outcomes in regard to heart failure. The results indicate that central sleep apnea is a significant risk factor to death and hospitalization compered to obstructive sleep apnea. This finding is potentially important in providing the optimal care to such patients.  

notes:

the percentage of central to total respiratory events (% C/T) should be better explain in the introduction.

 Where the death and hospitalization recorded related to cardio vascular reasons, if so pleas add it to the methods, if not pleas add it to the limitations of the study.

Author Response

We fully appreciate editor’s and 3 reviewers’ kind review and comments for our work. We attempted to incorporate them into the revised manuscript. We believe that the revision has been improved sufficiently enough for publication of your journal.

Following are responses to specific comments from the Reviewer 2.

Reviewer 2

In this study the researchers have examined whether different types of sleep apnea have different clinical outcomes in regard to heart failure. The results indicate that central sleep apnea is a significant risk factor to death and hospitalization compered to obstructive sleep apnea. This finding is potentially important in providing the optimal care to such patients. 

Specific comments:

the percentage of central to total respiratory events (% C/T) should be better explain in the introduction.

Response to the comment

The last sentence of the introduction was revised.

Where the death and hospitalization recorded related to cardio vascular reasons, if so pleas add it to the methods, if not pleas add it to the limitations of the study.

Response to the comment

The death in this study was defined as all-cause death. The hospitalization was hospitalization for HF. These changes were added in the revision.

Reviewer 3 Report

A sound and very correct article. I just wonder why you had to use data from almost 20 years ago; please add a phrase to motivate your choice. References are also quite old. Please find some more recent , 

Author Response

Response to Reviewers’ comments

We fully appreciate editor’s and 3 reviewers’ kind review and comments for our work. We attempted to incorporate them into the revised manuscript. We believe that the revision has been improved sufficiently enough for publication of your journal.

Following are responses to specific comments from the Reviewer 3.

Reviewer 3

A sound and very correct article. I just wonder why you had to use data from almost 20 years ago; please add a phrase to motivate your choice. References are also quite old. Please find some more recent

Response to the comment

We understand the point of the comment. We presented the results of this study at an International Congress in 2008. Then, we were going to make it published, but one coauthor who had managed the whole study were away from the study due to private concerns, which had hindered us to complete writing paper. However, one of the coauthors has suggested to publish the study because research on the association between frequency of central sleep apnea and clinical outcomes in patients with heart failure, have been scarce. Then, all authors have been motivated to complete this work for publication. References are updated. As indicated, references were updated.

Round 2

Reviewer 1 Report

The points I raised have been addressed satisfactorily, with the exception of points n. 5.

In lines 78-80 of the last version we can still read at the same time “Patients were not required to give informed consent” and “informed consent was obtained orally from all study participants”. Thus, in my view it is not clear whether informed consent was obtained or not.

Line 210. The term “positive” used in this line may be equivocal. Please, use a term that clarifies that CPAP reduced the risk.

Author Response

Response to Reviewers’ comments

We fully appreciate the reviewers’ kind review and comments for our work. We attempted to incorporate them into the second revised manuscript. We believe that the revision has been improved sufficiently enough for publication of your journal.

Following are responses to specific comments from Reviewers.

Reviewer 1

The points I raised have been addressed satisfactorily, with the exception of points n. 5.

Specific comments:

In lines 78-80 of the last version we can still read at the same time “Patients were not required to give informed consent” and “informed consent was obtained orally from all study participants”. Thus, in my view it is not clear whether informed consent was obtained or not.

Response to the comment

The sentences were revised as follows, “Informed consent was obtained orally from all study participants” (line 76 of the marked revision file).

Line 210. The term “positive” used in this line may be equivocal. Please, use a term that clarifies that CPAP reduced the risk.

Response to the comment

The part “However, CPAP therapy was proven to be a positive determinant for a composite of all-cause death and hospitalization for HF,” was revised to “However, CPAP therapy was proven to be associated with reduced risk for a composite of all-cause death and hospitalization for HF” (lines 201-203 of the marked revision file).
